# Sensitization of ZnO Photoconductivity in the Visible Range by Colloidal Cesium Lead Halide Nanocrystals

**DOI:** 10.3390/nano12234316

**Published:** 2022-12-05

**Authors:** Artem Chizhov, Marina Rumyantseva, Nikolay Khmelevsky, Andrey Grunin

**Affiliations:** 1Chemistry Department, Moscow State University, 119991 Moscow, Russia; 2Material Properties Research Laboratory (LISM), Moscow State Technological University Stankin,127055 Moscow, Russia; 3Faculty of Physics, Lomonosov Moscow State University, 119991 Moscow, Russia

**Keywords:** ZnO, perovskite nanocrystals, nanocomposites, sensitization, photoconductivity

## Abstract

In this work, colloidal perovskite nanocrystals (PNCs) are used to sensitize the photoconductivity of nanocrystalline ZnO films in the visible range. Nanocrystalline ZnO with a crystallite size of 12–16 nm was synthesized by precipitation of a zinc basic carbonate from an aqueous solution, followed by annealing at 300 °C. Perovskite oleic acid- and oleylamine-capped CsPbBr_3_, CsPb(Cl/Br)_3_ and CsPb(Br/I)_3_ PNCs with a size of 6–13 nm were synthesized by a hot injection method at 170 °C in 1-octadecene. Photoconductive nanocomposites were prepared by applying a hexane sol of PNCs to a thick (100 μm) polycrystalline conductive ZnO layer. The spectral dependence of the photoconductivity, the dependence of the photoconductivity on irradiation, and the relaxation of the photoconductivity of the obtained nanocomposites have been studied. Sensitization of ZnO by CsPbBr_3_ and CsPb(Cl/Br)_3_ PNCs leads to enhanced photoconductivity in the visible range, the maximum of which is observed at 460 and 500 nm, respectively; close to the absorption maximum of PNCs. Nanocomposites ZnO/CsPb(Br/I)_3_ turned out to be practically not photosensitive when irradiated with light in the visible range. The data obtained are discussed in terms of the position of the energy levels of ZnO and PNCs and the probable PNCs photodegradation. The structure, morphology, composition, and optical properties of the synthesized nanocrystals have also been studied by XRD, TEM, and XPS. The results can be applied to the creation of artificial neuromorphic systems in the visible optical range.

## 1. Introduction

Photoconductive structures based on wide-gap oxides, in particular, ZnO, are of great importance for the development of photodetectors [1,2,3] and gas sensors [4,5], ultraviolet lasers [6,7] and transparent conducting electrodes [8,9]. Recently, photoconductive structures attracted a lot of attention as promising elements of neuromorphic devices and in devices for image recognition based on artificial intelligence principles [10,11]. Thus, the use of photoconductive structures in the visible optical range is of interest in terms of the possibility of creating neuromorphic optical sensors. Zinc oxide is one of the most widely used semiconductors, exhibiting photosensitivity mainly to UV light (band gap 3.37 eV), although weak photoconductivity can also be observed in the visible range due to ionization of impurities or defects [12,13,14]. ZnO exhibits good environmental stability and high photosensitivity, besides being non-toxicity, eco-friendly, earthly abundance and low cost, so the development of ZnO-based photoconductors sensitive to visible light is of great importance. The effective sensitization of ZnO photoconductivity to visible light is a well-known challenge and is due to the need to develop photosensitive elements with a spectrally distributed sensitivity, the desire to use more economical radiation sources, or the possibility of using solar radiation energy [15].

One of the earliest approaches is the spectral sensitization of wide-gap oxides by organic dyes. It was shown that, for example, when ZnO is sensitized with eosin, methylene blue and crystal violet, the photoconductivity of the resulting structures upon irradiation with visible light increases up to $10^{4}$ times compared with the unsensitized sample [16,17,18]. Some modern studies also consider dye-sensitized metal oxides as promising functional materials [19,20,21,22]. However, when using dyes, a number of disadvantages are found, for example, low stability to heating and oxidation of dye molecules, which makes it difficult to manufacture structures with long-term stability. Also, dye molecules are prone to aggregation, as a result of which the formed aggregates have different optical properties compared to the properties of the solution of dyes, which makes it difficult to predict the optical properties of the sensitized structures. Another successful approach has been developed by using semiconductor quantum dots (QDs), such as CdSe, to sensitize the photoconductivity of wide-gap oxides [23,24,25,26,27,28,29,30]. The optical properties (i.e., absorption maximum) of QDs are varied by changing their size, thus, using the same material [31], it is possible to achieve photoconductivity sensitization in different ranges. Both in the case of dyes and quantum dots, the sensitization mechanism includes several stages, the absorption of a visible light by the sensitizer, the transition of the sensitizer to an excited state, and the injection of an electron from the excited level of the sensitizer into the conduction band of the metal oxide [32]. The possibility and rate of injection is determined by the total change in free energy as a result of this process, which includes several contributions: (i) the difference in energy between the excited level and the bottom of the conduction band of the metal oxide; (ii) the energy of formation of the charged state of the electron donor and acceptor and (iii) the energy of overcoming the Coulomb forces during the separation of the electron-hole pair [33].

Nanocrystals of the general composition APbX_3_, where A=CH_3_NH_3_^+^ (NH_2_)_2_CH^+^, Cs^+^ or Rb^+^; X=Cl^−^, Br^−^, or I^−^ with perovskite structure are new promising materials for optoelectronics due to the high luminescence quantum yield, high extinction coefficient, high carrier mobility [34]. Although perovskite nanocrystals (PNCs) also reveal size-dependent optical properties (in the size range less than two boron exciton radii), a more convenient way to tune the optical properties of halide perovskites is changing their anionic composition. Absorption edge of cesium-based perovskite nanocrystals covers the entire visible range from CsPbCl_3_ (405 nm) to CsPbBr_3_ (515 nm) and CsPbI_3_ (650 nm), while a continuous tuning in the optical properties of perovskite nanocrystals of mixed halide composition is possible [35].

The use of perovskite materials to improve the performance of photodetectors has been intensively studied recently. High performance perovskite-containing photosensitive elements operating in the mode of photodiodes, phototransitors and photoresistors have been demonstrated [36,37,38,39]. There is also an increasing number of studies considering photoconductive perovskite-containing materials as elements of artificial synapses [40,41,42,43,44]. The published results on the effect of perovskite nanocrystals on the photoconductivity of ZnO mainly concern monohalide perovskites with an organic [45,46,47,48,49] or inorganic cation [4,50,51,52,53,54,55,56,57], while the use of mixed halide perovskites as sensitizers use is also discussed in the literature [44,58,59,60], however has not been systematically studied so far.

In this work, we studied the effect of substitution of halogen atoms on the sensitizing properties of perovskite nanocrystals on the photoconductivity of ZnO. Three types of perovskite colloidal nanocrystals (PNCs) were synthesized: cesium lead bromide CsPbBr_3_, cesium lead bromide-chloride CsPb(Br/Cl)_3_ and cesium lead iodide-bromide CsPb(I/Br)_3_. The synthesized perovskite nanocrystals were applied to conducting layers of ZnO for the purpose of sensitization, and the spectral and photoconductive characteristics of the resulting nanocomposite structures were analyzed. The crystal structure, morphology, composition and optical properties of the synthesized nanocomposites are also presented.

## 2. Materials and Methods

### 2.1. Synthesis of Materials

#### 2.1.1. Synthesis of Nanocrystalline ZnO

Nanocrystalline ZnO was synthesized according to the method described in details in the our previous article [4] by the reaction between Zn(CH_3_COO)_2_ (Sigma Aldrich, Waltham, MA, USA, ACS reagent, ≥98%, #383058) and NH_4_HCO_3_ (BioUltra, ≥99.5%, #09830) in an aqueous medium. The resulting precipitate of zinc hydroxide carbonates was dried at 70 ${}^{\circ }$C and annealed at 300 ${}^{\circ }$C in air for 24 h.

#### 2.1.2. Synthesis of Perovskite Colloidal Nanocrystals

Colloidal PNCs were synthesized by hot-injection method in non-polar medium using Schlenk line according Protesescu et al. [35] with minimal variations (argon (99.998%, 7 ppm oxygen and 9 ppm water vapor) was used as an inert gas, the amounts of reagents were doubled compared to the original method). Three compositions of cesium lead halide nanocrystals were synthesized in this work: (1) pure bromide; (2) mixed chloride-bromide with a loaded molar ratio of Cl:Br = 1:2; (3) mixed bromide-iodide with a loaded molar ratio of I:Br = 1:2. Below, mixed-halide nanocrystals will be signed as CsPb(Br/Cl)_3_ and CsPb(I/Br)_3_, and the clarifying of chemical composition of synthesized PNCs will be given in the Section 3.1.4.

In the 3-neck 25-mL flask was poured 10 mL 1-octadecene (ODE, Sigma Aldrich, 90%, #O806), than was added 0.376 mmol PbBr_2_ (Sigma Aldrich, 99.999%, #398853) to synthesize the CsPbBr_3_ PNCs; or a mix of 0.251 mmol PbBr_2_ and 0.125 mmol PbCl_2_ (Sigma Aldrich, 99.999%, #203572) to synthesize the CsPb(Br/Cl)_3_ PNCs; or a mix of 0.251 mmol PbBr_2_ and 0.125 mmol PbI_2_ (Alfa Aesar, Haverhill, MA, USA, 99.999%, ultra dry, #44314) to synthesize the CsPb(I/Br)_3_ PNCs. Further, the synthesis protocol was the same in the synthesis of three types of nanocrystals. The reaction mixture was left under vacuum for 30 min at a temperature of 120 ${}^{\circ }$C and vigorous stirring. After degassing procedure, 1 mL of oleylamine (Sigma Aldrich, 70%, #O7805) and 1 mL of oleic acid (OA, Sigma Aldrich, 90%, #364525) were injected in the flask and degassing procedure was performed again. When lead halides were fully solubilized, the temperature of reaction mixture was raised to 170 ${}^{\circ }$C and 0.8 mL of 0.125 M warm cesium oleate solution in ODE was injected in the flask under Ar. After completion of the reaction (5 s), the flask was cooled to room temperature, the reaction mixture was poured into centrifuge tubes and cooled to 0 ${}^{\circ }$ for a more complete isolation of NCs. Centrifugation was carried out at 10,000 rpm for 10 min. The resulting precipitate of PNCs was separated as far as possible from traces of ODE and stored as such.

A solution of cesium oleate was previously prepared by reacting 1.25 mmol of Cs_2_CO_3_ (Alfa Aesar, Puratronic^®^, 99.994%, #12117) with 1.25 mL of OA in 20 mL of ODE by heating to 120 ${}^{\circ }$C under vacuum.

#### 2.1.3. Fabrication of Photoconductive Elements

Conductive layers were formed on Al_2_O_3_ substrates 1.5 × 1.5 mm in size equipped with Pt measuring contacts with a distance of 200 μm between them. The synthesized nanocrystalline ZnO (5 mg) was mixed with 50 μL of α-terpineol in an agate mortar and the resulting mixture was ground to form a thick dispersion, which was applied to the surface of the measuring plate with a volume of 2–3 μL, then dried at 70 ${}^{\circ }$C and annealed at 300 ${}^{\circ }$C for 24 h. After annealing, a nanocrystalline conductive ZnO film with a thickness of about 100 μm with a linear current-voltage characteristic was obtained. A more detailed characterization of the ZnO layers obtained by this method is given in our previous works, for example [61].

For sensitization, the synthesized PNCs was dispersed in *n*-hexane to a concentration of about 10 mg/mL. Approximately 2–3 μL of a freshly prepared dispersion of PNCs was dropped onto the surface of the ZnO conducting layer and than dried for 60 min at 70 ${}^{\circ }$C. A schematic representation of the structure of a photosensitive element based on a ZnO/PNCs nanocomposite is shown in Figure 1. Images of photoconductive ZnO layers sensitized by three types of PNCs are shown in the Figure 2. The content of PNCs in the resulting ZnO/PNCs nanocomposites was controlled by X-ray fluorescence analysis [62]; from the set of fabricated photosensitive elements only some with a close content of nanocrystals of 5±1% were selected for experiments.

### 2.2. Characterization of Materials

Phase composition and crystal structure of synthesized materials was studied by powder X-ray diffraction (XRD) with a Rigaku diffractometer (Rigaku Corporation, Tokyo, Japan) using ${\mathrm{CuK}}_{\alpha }$ radiation (wavelength λ = 1.54059 Å). Average crystallite size *D* was calculated using the Sherrer equation:(1)$$D=\frac{k\lambda }{\sqrt{\beta_{exp}^{2}-\beta_{app}^{2}}\cos\Theta }$$
where λ is a wavelength of X-ray radiation, nm; $\beta_{exp}$ is the observed peak width at half height and $\beta_{app}$ is the instrumental broadening, rad; θ is a diffraction angle; *k* is a coefficient equal to 0.9.

Diffuse reflectance spectra were recorded in the wavelength range of 300–800 nm on a Perkin–Elmer Lambda 35 spectrophotometer (PerkinElmer, Inc., Waltham, MA, USA). Absorption spectra were recalculated using the Kubelka–Munk function (F) according equation
(2)$$F=\frac{{\left(1-R_{d}\right)}^{2}}{2R_{d}}$$
where $R_{d}$ is the diffuse reflectance coefficient.

The absorption spectra of liquid samples were recorded using a Varian Cary 50 spectrophotometer (Agilent Technologies, Santa Clara, CA, USA) in the range of 300–1000 nm. The photoluminescence (PL) spectra of PNCs dispersions were recorded using a USB-4000 spectrometer (Ocean Insight, Orlando, FL, USA) and a 405 nm laser as the excitation source. X-ray photoelectron spectroscopy (XPS) measurements were performed using a K-Alpha spectrometer (Thermo Scientific, Prague, Czech Republic) with an Al Kα X-ray source (E=1486.7 eV). The main state of C1s core level was used as a reference with a binding energy (BE) of 285 eV. The morphology and size of colloidal PNCs were studied using a LEO 912 AB Omega (Zeiss, Oberkochen, Germany) transmission electron microscope (TEM).

### 2.3. Photoconductivity Measurements

The spectral dependence of the photoconductivity of the nanocomposites was measured using an optical setup consisting of a radiation source (100 W halogen lamp), a condenser and a monochromator MDR-41 (“OKB Spectr”, St. Petersburg, Russia) based on a diffraction grating (1500 lines/mm, spectral range 410–780 nm). The width of the entrance slit of the monochromator was set to 0.2 mm, resulting in a spectral line FWHM of 1.2 nm. A photoconductive element based on nanocomposites was fixed opposite the exit slit of a monochromator equipped with a focusing lens. The electrical resistance of the samples was measured by a two-probe method using Keithley 6517A (Tektronix, Beaverton, OR, USA) at 1 V bias. Under dark conditions, we waited for the establishment of dark conductance $\sigma_{dark}$, which was about 1 GΩ. After that, the photosensitive layer was irradiated with monochromatic light for 10 s, and the obtained conductance was measured ($\sigma_{light}$). Before each subsequent measurement, the conductivity of sample was allowed to relax to the dark value. Relative photoconductivity was calculated by the formula:(3)$$\Phi =\frac{\sigma_{light}-\sigma_{dark}}{\sigma_{dark}}$$

Raw photoconductivity spectra (Φ(λ)) were normalized to photon flux to yield a normalized spectrum of photoconductivity ($\Phi_{N}\left(\lambda \right)$). Irradiance of monochromatic light ($W/{\mathrm{cm}}^{2}$) was measured using Nova II radiometer (Ophir, Jerusalem, Israel) equipped with photodiode PD300-UV-193 head.

The study of kinetics and stationary photoconductivity was carried out on a laboratory-made setup that allows recording the electrical resistance of photoconductive elements in the range from $10^{-1}$ to $10^{10}$ Ω with a discretization of 0.05 s. The photoconductive elements were fixed in a gas- and lightproof Teflon cell. A LED ($\lambda_{max}=470$ nm) inside the cell was used for irradiation. The distance from the LED to the photoconductive elements was approximately 4 cm. The measurements were carried out in dry air at room temperature.

## 3. Results

### 3.1. Characterization of Nanocrystals

#### 3.1.1. Crystal Structure and Morphology

The diffraction pattern of synthesized nanocrystalline ZnO is shown in Figure 3a. All the observed reflections belong to the ZnO phase with the wurtzite structure. The average crystallite size calculated by the Scherrer Formula (Equation 1) is 12–16 nm. An exhaustive characterization of nanocrystalline ZnO synthesized by the same method, including morphology, atomic charge states, was given in our previous work [4], so in what follows we will focus on the characterization of synthesized PNCs.

The XRD patterns of the synthesized PNCs are shown in Figure 3b. As is known, halide PNCs can have a cubic, tetragonal or orthorhombic crystal structure [63,64]. Routine XRD analysis does not make it possible to establish the exact crystal structure of the synthesized PNCs. However, it can be argued that the synthesized PNCs have a perovskite structural type. For example, the obtained reflections of CsPbBr_3_ PNCs are compared with the standard diffraction pattern of a cubic structure (00-054-0752, PDF4). The observed reflections can completely correspond to the cubic phase of CsPbBr_3_, ($Pm\bar{3}m$ space group). When a part of bromine atoms in CsPbBr_3_ NCs is substituted by chlorine atoms, X-ray reflections shift towards larger 2θ angles; on the contrary, when bromine atoms is substituted by iodine atoms, the X-ray reflections shift towards smaller 2θ angles. Thus, the substitution of Br atoms of smaller atoms (Cl) leads to a decrease in the unit cell parameters of the resulting NCs, while the substitution of Br of larger atoms (I) leads to an increase in the unit cell parameters.

The synthesized PNCs have a cubic morphology or rectangular parallelepipeds (Figure 4a,d,g). Despite the fact that the nanocrystals were synthesized under the same conditions, their average sizes differ significantly: 6.1±1.3 nm for CsPb(Br/Cl)_3_, 8.9±2.5 nm for CsPbBr_3_, and 13.2±5.3 nm for CsPb(I/Br)_3_. For all three types of PNCs, the statistical size distribution is most likely lognormal. Thus, in the synthesis of mixed chloride-bromide PNCs, the size obtained is smaller than in the synthesis of pure bromide nanocrystals under the same experimental conditions; the polydispersity of PNCs also decreases. An opposite trend is observed in the synthesis of mixed iodide-bromide PNCs: their size is larger compared to pure bromide PNCs and polydispersity also increases (Figure 4b,e,h). An increase in polydispersity in the case of iodine-containing PNCs may be due to a higher rate of the reaction of formation of nanoparticles, as a result of which the growth of nanoparticles begins immediately after the injection of cesium oleate, even before the full mixing of precursors. On the other hand, during the synthesis of chlorine-containing PNCs, the growth of nanocrystals is slowed down and begins a few seconds after the injection of cesium oleate. Thus, the precursors well mixed and conditions are created for uniform supersaturation throughout the volume of the reaction mixture, resulting in the formation of monodisperse nanoparticles. The electron diffraction patterns also confirm the perovskite structure of the investigated PNCs (Figure 4c,f,i).

#### 3.1.2. Optical Properties

The absorption spectra of PNCs dispersions in hexane are shown in the Figure 5. Each of the samples is characterized by an absorption edge in the visible region, while the substitution of bromine with chlorine atoms leads to a shift in the absorption edge to the short-wavelength region, and upon substitution with iodine, on the contrary, to the long-wavelength region. PNCs dispersions also exhibit single-peak photoluminescence, the band of which also undergoes a short- or long wavelength shift according to the type of atoms being substituted. So, for CsPbBr_3_ PNCs maximum of PL located at 517 nm, for CsPb(Br/Cl)_3_ at 483 nm and for CsPb(I/Br)_3_ at 542 nm. The calculated values of the band gap of PNCs using the Tauc plot assuming of direct allowed transitions are shown in the inset to the Figure 5.

#### 3.1.3. Charge States of Atoms in PNCs

XP spectra of the synthesized PNCs are shown in Figure 6. The presence of all elements in the samples corresponding to the loaded composition of the PNCs was confirmed by XPS. The position of photoelectronic peaks on the binding energy scale is given in the Table 1.

The Cs3d doublet in CsPbBr_3_ has a single charge state ($\mathrm{Cs}3d_{5/2}\left(I\right)$, 724.4 eV) relates to Cs^+^ ions in the perovskite lattice. Lead in CsPbBr_3_ PNCs exhibits two charge states, one of which, with a higher intensity ($\mathrm{Pb}4f_{7/2}\left(I\right)$, 138.4 eV), refers to Pb^2+^ ions in the perovskite lattice and the state with a lower intensity ($\mathrm{Pb}4f_{7/2}\left(\mathrm{II}\right)$, 136.9 eV) can presumably be attributed to metallic lead. The intensity ratio of states (I) and (II) is approximately 8.6:1. Traces of metallic Pb in perovskites founded by XPS was noted in previous works and is probably a consequence of the insignificant photolysis of nanocrystals during the analysis [65]. Bromine in CsPbBr_3_ is represented by a Br3d doublet (spin orbital splitting 1.0 eV) with a single charge state ($\mathrm{Br}3d_{5/2}$(I), 68.5 eV), which relates to bromine ions in the perovskite lattice.

The charge states of the Cs, Pb, Br, elements in CsPb(Br/Cl)_3_ PNCs are similar to those considered earlier for CsPbBr_3_, with the only difference that a higher intensity of the metallic lead signal is observed, the ratio of the intensities of states (I) and (II) is approximately 3.9:1. Although bromine in CsPb(Br/Cl)_3_ sample also has a single charge state, however, the FWHM of the $\mathrm{Br}3d_{5/2}$ and $\mathrm{Br}3d_{3/2}$ peaks is increased and the peaks are not resolved. Apparently, this is due to the fact that the nearest environment of bromine atoms in the chlorine-substituted perovskite lattice may have a different proportion of bromine and chlorine atoms around it, which leads to a statistical distribution of the charge states of Br atoms in the perovskite lattice and broadening of spectral lines. The Cl2p doublet shows a single charge state with a peak $\mathrm{Cl}2p_{3/2}$ located at 197.9 eV, which corresponds to the charge state of chloride ions.

The charge states of elements in CsPb(I/Br)_3_ differ significantly from the previous two cases considered. First, two charge states of cesium are observed, which may be evidence of the presence of Cs^+^ both in the composition of the perovskite lattice and in another phase, possibly cesium halide. For lead, two charge states are also observed, moreover, the characteristic state of metallic lead is absent and instead, on the contrary, a state with a higher binding energy is observed, compared with perovskite lattice lead. For bromine and iodine, the splitting of doublets into two charge states with different intensity is also noted. Thus, the XP spectra obtained indicate partial decomposition of the perovskite phase on the surface of the CsPb(I/Br)_3_ nanoparticles with formation of cesium and lead halides.

#### 3.1.4. Chemical Composition of PNCs

The surface chemical composition of the PNCs was calculated from the XP spectra presented on the Figure 6. The calculated composition in atomic percent is given in the Table 2. For CsPbBr_3_ PNCs, a close to stoichiometric ratio of elements is observed. Considering the halogen content to be integer, there is a slight lack of cesium in the composition (Cs_0.90_Pb_1.03_Br_3_). In CsPb(Br/Cl)_3_ sample, the calculated ratio between bromine and chlorine is 2.3:1, slightly more than the expected 2:1. In this case, the total content of halogen atoms is approximately equal to the atomic fraction of lead in the sample, but for cesium there is also a slight deficiency and the resulting composition corresponds to the formula Cs_0.7_Pb_1.2_Br_2.1_Cl_0.9_. In CsPb(I/Br)_3_, the total ratio of Br:I (without taking into account their charge state) is 5:1, which differs significantly from the loaded ratio of 2:1. The overall composition of the synthesized PNCs is Cs_0.7_PbI_0.5_Br_2.5_, and as for the two previous samples, there is a deficiency of cesium is observed.

The composition of PNCs, synthesized by the same procedure as those discussed in this article, was previously studied by us using inductively coupled plasma mass spectrometry (ICP-MS) and total reflection X-ray fluorescence (TXRF) spectroscopy [62]. Using precise analytical techniques, the stoichiometric composition of cesium lead bromide PNCs was confirmed (CsPbBr_3_), and compositions CsPbBr_2_Cl and CsPbI_0.3_Br_2.7_ were obtained for mixed-halide PNCs. Thus, the composition of PNCs determined by XPS is generally in agreement with the composition determined by analytical methods. It is interesting to note that the surface composition of CsPb(Br/Cl)_3_ PNCs shows a slight lower Cl/Br ratio than in the overall composition of this PNCs. In contrast, CsPb(I/Br)_3_ PNCs shows a higher I/Br ratio at the surface compared the overall composition. Thus, with halogen substitution, chlorine atoms are more likely to enter the bulk of nanocrystals, while iodine atoms, on the contrary, concentrate near the surface. This fact finds an explanation in accordance with the Goldsmith’s rule, according to which the substitution of bromine by chlorine in CsPbBr_3_ leads to an increase in the tolerance factor (its tendency to 1), which means an increase in the stabilization of the perovskite structure; on the contrary, upon substitution with iodine, the tolerance factor decreases, which leads to a decrease in the stability of the perovskite structure and, therefore, iodine does not easily substitute bromine in the perovskite lattice and concentrates near the PNCs surface.

In general, it was found that the composition of synthesized mixed chloride-bromide PNCs is as close as possible to the loaded composition (Br:Cl = 2:1), while mixed iodide-bromide PNCs significantly deviate in their loaded composition (Br:I = 2:1) and demonstrate a significant deficiency of iodine.

### 3.2. Characterization of ZnO/PNCs Nanocomposites

The study of the microstructure of ZnO/PNCs nanocomposites by electron microscopy is presented in our previous work [4] using ZnO/CsPbBr_3_ nanocomposite an example. It was shown that individual oleic acide- oleylamine-capped CsPbBr_3_ NCs are attached on the surface of ZnO crystallites without a significant change in morphology. In this work, we restrict ourselves to examining the phase composition and optical properties of synthesized ZnO/PNCs nanocomposites with a PNCs content of 5 ± 1%.

XRD patterns of ZnO/PNCs nanocomposites are shown in Figure 7 (intensity is plotted on a logarithmic scale). The presented diffraction patterns show both peaks corresponding to the ZnO phase with the wurtzite structure and peaks corresponding to the PNCs with the perovskite structure. Compared to the diffraction patterns of individual PNCs (Figure 3b), the position of the peaks in the nanocomposites did not changed, and no new peaks appeared either, which indicates that the phase composition of the components was retained as a result of the formation of the nanocomposite.

The absorption spectra of the ZnO/PNCs nanocomposites are discussed in the following Section 3.3 and Section 4. Freshly prepared synthesized ZnO/PNCs nanocomposites exhibit an absorption edge in the visible region, which is located near the absorption edge of the corresponding PNCs and is shifted relative to it by 5–7 nm towards shorter wavelengths.

### 3.3. Photoconductivity of ZnO/PNCs Nanocomposites

The synthesized ZnO/CsPb(Br/Cl)_3_ and ZnO/CsPbBr_3_ nanocomposites showed high photosensitivity, decreasing their electrical resistance under visible irradiation, while ZnO/CsPb(I/Br)_3_ nanocomposite shows much lower photosensitivity, only slightly changing its resistance when irradiated with a high-power LED or laser beam.

Thus, the Figure 8a shows the photoresponse of ZnO/PNCs nanocomposites and unsensitized nanocrystalline ZnO upon irradiation with a blue LED ($\lambda_{max}=470$ nm, 13 $\mathrm{mW}/{\mathrm{cm}}^{2}$) for 30 s at bias 4 V. The selected radiation wavelength lies in the absorption region of all three nanocomposites, therefore, the photoresponse was also expected for all three nanocomposites. It can be seen from the Figure 8a, all studied samples (ZnO/PNCs nanocomposites and non-sensitized ZnO) demonstrate a photoresponse, but of a different order. The most intense photoresponse is observed for ZnO/CsPb(Br/Cl)_3_ and ZnO/CsPbBr_3_ nanocomposites respectively, but the ZnO/CsPb(I/Br)_3_ nanocomposite demonstrates phtotoresponce comparable to the non-sensitized ZnO. Photosensitivity of non-sensitized ZnO in this spectral range can be result of the photoexcitation of electrons from impurity and defect levels lying within the bandgap. The sensitization with PNCs also affects the dark conductivity; in the case of ZnO/CsPb(Br/Cl)_3_ and ZnO/CsPbBr_3_ nanocomposites, it increases by about an order of compared to unsensitized ZnO, and for ZnO/CsPb(I/Br)_3_ nanocomposite, it is approximately equal to the dark conductivity of the ZnO. The responsivity (R) in the photoresistance mode, calculated as the ratio of the photocurrent to the power of the incident radiation, was 20 and 16 mA/W for ZnO/CsPb(Br/Cl)_3_ and ZnO/CsPbBr_3_ nanocomposites respectively, and about 20–40 times less for ZnO/CsPb(I/Br)_3_ and non-sensitized ZnO. The main photoelectric parameters of ZnO/PNCs nanocomposites and non-sensitized ZnO under blue LED irradiation (Figure 8), including dark conductance, on/off ratio, responsivity and photoconductivity decay time by 90% are presented in the Table 3.

Attention is drawn to the difference in the kinetics of the rise in the photoconductivity of the studied samples. ZnO/PNCs nanocomposites show a sharp transition in high photoconductive state at the moment the light is turned on, and then, during the irradiation period, the photoconductivity decreases. In contrast, unsensitized ZnO exhibits a continuous increase in photoconductivity within the irradiation period. Observed kinetics of rise in photoconductivity in the case of nanocomposites can be explained from the sensitization mechanism. Under the light irradiation, nonequilibrium charge carriers are generated in PNCs; photoexcited electrons are injected into the conduction band of ZnO, and photoexcited holes remain in PNCs. Since the recombination of photoexcited holes is hindered, PNCS under irradiation acquire a positive charge and an opposite gradient of the electric field arises, which slows down the rate of electron injection. As result, the concentration of charge carriers in ZnO reduces and photocurrent also decreases. In the case of ZnO, the kinetics of rise of photoconductivity has a classical form due to competing processes of generation and recombination of photoexcited charge carriers in a single-phase semiconductor.

Based on the revealed difference in kinetics, it can be assumed that in the case of ZnO/CsPb(I/Br)_3_ nanocomposite, photoinduced electron transfer still takes place, although at a very low rate. In order to more reliably investigate this process, we irradiated ZnO/CsPb(I/Br)_3_ nanocomposite and non-sentisized ZnO with light in the absorption band of CsPb(I/Br)_3_ nanocrystals (525 nm, 13 $\mathrm{mW}/{\mathrm{cm}}^{2}$). As shown on the Figure 8b, the impurity photoconductivity of ZnO when irradiated with green light is expected less than when irradiated with blue light, while the photoconductivity of ZnO/CsPb(I/Br)_3_ nanocomposite, on the contrary, increases, which directly confirms the sensitization of ZnO photoconductivity by CsPb(I/Br)_3_ PNCs, although with a very low efficiency. So, a clear difference in the kinetics of the rise in photoconductivity between the ZnO/CsPb(I/Br)_3_ nanocomposite and the non-sensitized ZnO under green LED irradiation is also observed.

The Figure 9 shows the spectral dependences of the photoconductivity of ZnO/CsPbBr_3_ and ZnO/CsPb(Br/Cl)_3_ nanocomposites, normalized to the photon flux, in the range of 410–550 nm under irradiance of 5–10 $\mu W/{\mathrm{cm}}^{2}$ at room temperature in comparison with the absorption spectra of ZnO/PNCs nanocomposites and PNCs dispersions. Both nanocomposites exhibit an edge of increasing photoconductivity, coinciding with the position of the absorption edge of the corresponding PNCs, which passes into a peak with a maximum at 460–465 nm for ZnO/CsPb(Br/Cl)_3_ nanocomposite and with a maximum at 495–500 nm for ZnO/CsPbBr_3_ nanocomposite. It can be seen from the Figure 9 that the edge of the absorption spectrum of the ZnO/PNCs nanocomposites is shifted by approximately 5 nm to the blue region relative to the absorption spectrum of the corresponding PNCs dispersion in hexane; however, the edge of increasing photoconductivity follows rather the absorption spectrum of the PNCs dispersion than the nanocomposite’s absorption spectrum. As the photon energy increases, the photoconductivity of nanocomposites somewhat decreases, but remains high up to 410 nm. The spectral dependence of the photoconductivity of ZnO/CsPb(I/Br)_3_ nanocomposite could not be registered due to low photosensitivity and high resistance. The photoconductivity of this nanocomposite was not observed even under direct irradiation with light with the maximum absorption wavelength of nanocrystals (530 nm) and irradiance up to 100 $\mu W/{\mathrm{cm}}^{2}$.

The dependence of the photoconductivity of ZnO/CsPb(Br/Cl)_3_ and ZnO/CsPbBr_3_ nanocomposites on irradiance at room temperature is shown in Figure 10, when irradiated with light with a wavelength corresponding to the maximum photosensitivity (460 nm for ZnO/CsPb(Br/Cl)_3_ and 500 nm for ZnO/CsPbBr_3_), which was previously determined from the spectral dependences of photoconductivity on the Figure 9. In general, the obtained dependences are linearized in semilogarithmic coordinates; those, the dependence of relative photoconductivity on irradiance has an exponential form:(4)Φ=Aexp(βI)
where *I*—irradiance. For the ZnO/CsPb(Br/Cl)_3_ nanocomposite, the dependence has a steeper slope and is characterized by a coefficient β=0.4, while for nanocomposite ZnO/CsPbBr_3_, a more flat dependence is observed with the corresponding coefficient β=0.2. The minimum irradiance at which the nanocomposites exhibited photosensitivity was about 0.6 $\mu W/{\mathrm{cm}}^{2}$ for ZnO/CsPb(Br/Cl)_3_ and 4.0 $\mu W/{\mathrm{cm}}^{2}$ for ZnO/CsPbBr_3_. At the same time, the minimum irradiance at which the non-sensitized ZnO demonstrates photosensitivity was found to be about 100 $\mu W/{\mathrm{cm}}^{2}$ at 380 nm, thus PNCs significantly enhance the photosensitivity of the resulting ZnO/PNCs nanocomposites. The lower sensitivity of unsensitized ZnO to irradiation can be associated with a high exciton binding energy in this material (60 meV), as a result of which excitons are the first to form upon absorption of light, and free carriers appear upon further exciton dissociation. In the case of ZnO/PNCs nanocomposites, direct injection of electrons from PNCs occurs without the formation of excitons in ZnO, which causes higher photosensitivity and a sharp photoconductivity peak in the initial period of irradiation.

## 4. Discussion

Despite the promising applying of ZnO in opto- and photoelectronic devices, for photoactivated gas sensing and other applications, the use of pure ZnO is limited by the UV spectral range in which it exhibits photoconductivity. Sensitization by substances that intensely absorb light in the visible range makes it possible to expand the spectral sensitivity range of ZnO and shift it towards lower photon energies, however, this requires the development of new models of photosensitivity, taking into account the increase in the physical and chemical complexity of the photosensitive material.

Photoinduced transfer of photoexcited electrons from sensitizer particles to ZnO grains is due, in general, to a decrease in the free energy of the system. In the first approximation, the possibility of this process is determined by the difference in the position of the excited energy level of sensitizers (perovskite nanocrystals) and ZnO; for photoinduced transfer, $E_{c}\left(\mathrm{ZnO}\right)<E_{c}\left(\mathrm{PNCs}\right)$ is necessary. Figure 11 shows a comparative band diagram of the position of the band edges of ZnO and PNCs [66]. As can be seen, condition $E_{c}\left(\mathrm{ZnO}\right)<E_{c}\left(\mathrm{PNCs}\right)$ is satisfied for all synthesized PNCs, and photoinduced electron injection is possible. If we consider CsPbBr_3_ PNCs as a starting point, the substitution of Br atoms with Cl leads, firstly, to an increase in the band gap due to a slight increase in the energy $E_{c}$ (+0.05 eV) and a more significant decrease in $E_{v}$. When Br atoms in CsPbBr_3_ PNCs is substituted by I, the opposite trend occurs, the decreasing of the band gap due to a slight decrease in $E_{c}$ (−0.05 eV) and a more significant increase in $E_{v}$.

Thus, the halide substitution in PNCs has little effect on the energy position of $E_{c}$, while the energy difference $E_{c}\left(\mathrm{PNCs}\right)-E_{c}\left(\mathrm{ZnO}\right)$ for the synthesized nanocrystals is about 1 eV, so it can be expected that all three synthesized nanocomposites will exhibit high photosensitivity when irradiated with visible light. This is confirmed in the case of nanocomposites ZnO/CsPbBr_3_ and ZnO/CsPb(Cl/Br)_3_; however, ZnO/CsPb(Br/I)_3_ nanocomposite, as noted above, has an extremely low photosensitivity. This can be explained by the assumption that, under ambient conditions, iodine-containing PNCs have low stability and are subject to moisture- and light-induced degradation. One of the confirmations for this can be found in the XP spectra (Figure 6), which indicate a partial surface decomposition of the perovskite phase of CsPb(Br/I)_3_ PNCs. On the other hand, X-ray diffraction does not reveal foreign crystalline phases in the ZnO/CsPb(Br/I)_3_ nanocomposite, which can also indicate only the surface decomposition process in this PNCs, as a result of which a dielectric shell is formed on their surface, which creates an additional potential barrier between the CsPb(Br/I)_3_ PNCs and ZnO grains and hinders the photoinduced electron transfer.

To evaluate the photostability of the obtained nanocomposites, we performed an additional experiment by irradiating all three freshly prepared nanocomposites with high-power blue light (470 nm, 40 $\mathrm{mW}/{\mathrm{cm}}^{2}$) for 48 h and controlling their optical properties before and after irradiation (Figure 12). It can be seen that ZnO/CsPb(Br/I)_3_ nanocomposite significantly changed its optical properties after irradiation, the absorption edge shifted by about 20 nm towards shorter wavelengths; the spectrum obtained after irradiation is very close to the absorption spectrum of ZnO/CsPbBr_3_ nanocomposite, from which it can be assumed that the photodegradation of iodine-containing perovskites occurs with the formation of CsPbBr_3_ phase. ZnO/CsPb(Cl/Br)_3_ also changed its properties after irradiation, the initial absorption edge changed the slope and the absorption maximum shifted by about 30 nm towards shorter wavelengths. The results obtained also demonstrate the potential instability of the mixed chloride-bromide perovskite phases under intense light irradiation, although the photoelectric characteristics of ZnO/CsPb(Cl/Br)_3_ nanocomposite showed good stability when irradiated with light of low intensity (microwatt range). Finally, the monohalide ZnO/CsPbBr_3_ nanocomposite practically did not change the optical properties after irradiation and thus demonstrated better photostability.

The carried out studies of stationary photoconductivity, spectral and irradiation dependence of the photoconductivity showed that ZnO/CsPb(Cl/Br)_3_ nanocomposite demonstrates slightly better photoelectric parameters then ZnO/CsPbBr_3_, for example, in sensitivity to radiation, and also in responsivity. The reasons for this may be different. Firstly, the position of the $E_{c}$ level in CsPb(Cl/Br)_3_ is somewhat higher than in CsPbBr_3_ PNCs (by 0.05 eV), which, theoretically, can lead to a larger difference in energy between $E_{c}\left(\mathrm{ZnO}\right)$ and $E_{c}\left(\mathrm{PNCs}\right)$; although, based on the calculated data, such a small difference in energy can hardly lead to a significant increase in the electron injection rate. Another explanation is the effect of the size of PNCs on the efficiency of sensitization. Indeed, CsPb(Cl/Br)_3_ PNCs synthesized in this work are almost 1.5 times smaller than CsPbBr_3_ PNCs; therefore, at the same mass loading in nanocomposites, CsPb(Cl/Br)_3_ PNCs will create a higher surface density of sensitization centers, which will also lead to an increase in the electron injection rate.

## 5. Conclusions

Summarizing the above, in this work we have shown the possibility of sensitization of the photoconductivity of nanocrystalline ZnO by colloidal perovskite nanocrystals. Although available chemical methods make it possible to synthesize PNCs with optical absorption covering the entire optical range, not all of them are equally effective for photoconductivity sensitization. Bromide and mixed chloride-bromide cesium lead halides effectively sensitize the photoconductivity of ZnO in the spectral range of 400–500 nm, while mixed iodine-bromine PNCs proved to be weak sensitizers in the corresponding spectral range of more than 500 nm. The reason for this may lie in the possible photo- and moisure-induced degradation of CsPb(I/Br)_3_ nanocrystals. The synthesized ZnO/CsPbBr_3_ and ZnO/CsPb(Br/Cl)_3_ nanocomposites exhibited a clear photoconductivity edge coinciding with their absorption spectra and significantly (by 2 orders of magnitude) increasing the sensitivity to radiation compared by non-synthesized ZnO. ZnO/PNCs nanocomposites demonstrated remarkable photoconductivity kinetics, since their photoconductivity did not rise, but decay during irradiation, which can be used to regulate synaptic plasticity. Thus, the studied ZnO/PNCs nanocomposites can be discussed as potential promising materials for photo- and optoelectric, photoelectric, sensors applications and open up new possibilities for neuromorphic applications, in particular to control the response of artificial optoelectronic synapses.

## Figures and Tables

**Figure 1 nanomaterials-12-04316-f001:**
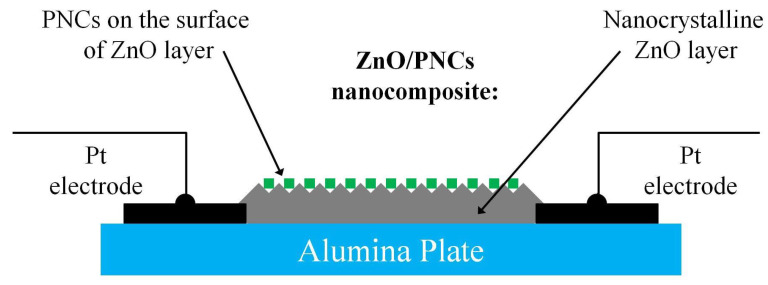
Schematic representation of the structure of a ZnO/PNCs photosensitive element, including alumina plates with Pt contacts, a nanocrystalline ZnO thick layer, and PNCs covering the top of ZnO layer.

**Figure 2 nanomaterials-12-04316-f002:**
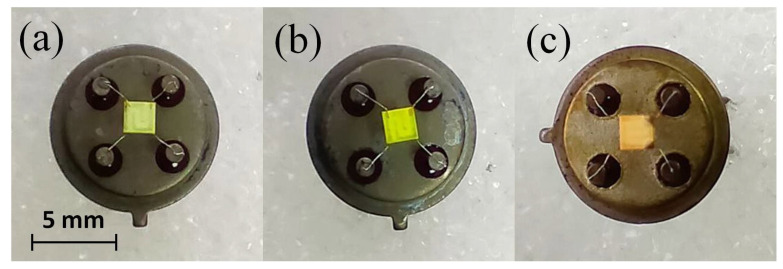
Images of ZnO/CsPb(Br/Cl)_3_ (**a**), ZnO/CsPbBr_3_ (**b**) and ZnO/CsPb(I/Br)_3_ (**c**) photoconductive nanocomposites formed on the measuring aluminia plates with Pt contacts.

**Figure 3 nanomaterials-12-04316-f003:**
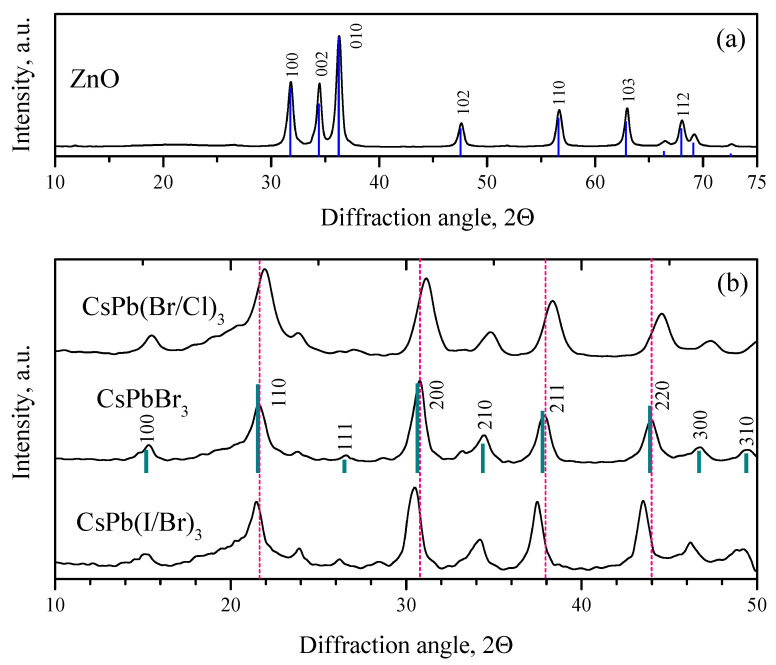
XRD patterns of nanocrystalline ZnO (**a**) and perovskite nanocrystals (**b**). The bar diffraction pattern corresponds to the reflections of the standard diffraction pattern of the cubic phase CsPbBr_3_ (00-054-0752, PDF4).

**Figure 4 nanomaterials-12-04316-f004:**
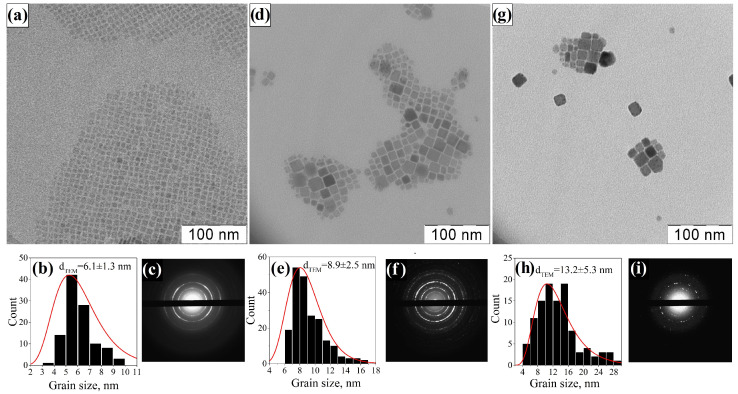
TEM images, size distribution diagrams and electron diffraction patterns of CsPb(Br/Cl)_3_ (**a**–**c**); CsPbBr_3_ (**d**–**f**); and CsPb(I/Br)_3_ (**g**–**i**).

**Figure 5 nanomaterials-12-04316-f005:**
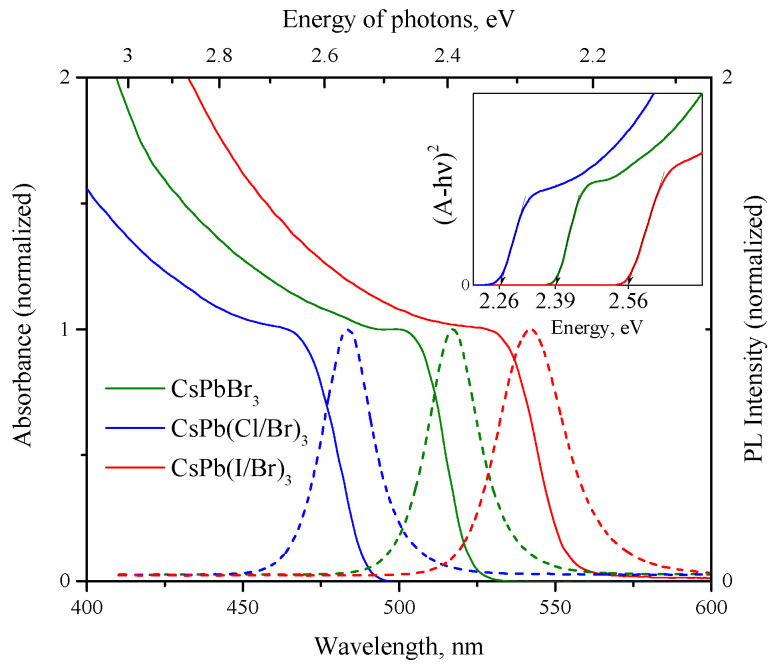
Absorbance (solid lines) and PL (dashed lines) spectra of PNCs hexane dispersions. On the inset shown determination of optical band gaps of PNCs using Tauc plot.

**Figure 6 nanomaterials-12-04316-f006:**
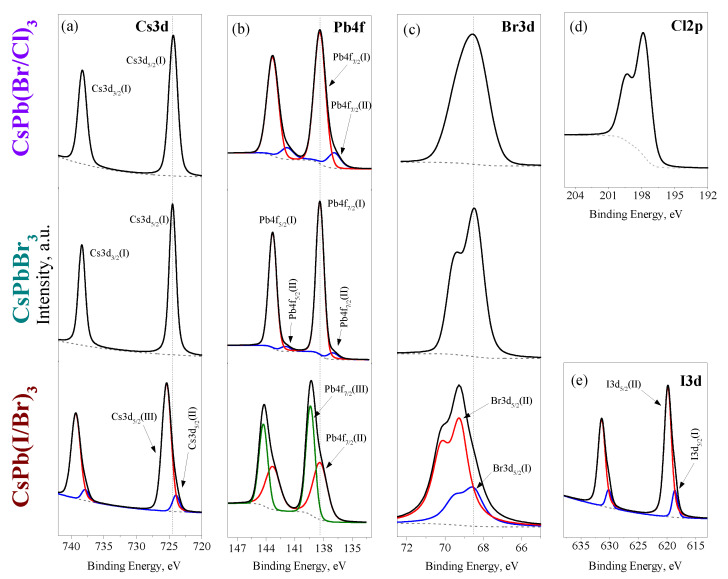
XP spectra of PNCs on the Cs3d (**a**), Pb4f (**b**), Br3d (**c**), Cl2p (**d**) and I3d (**e**) regions.

**Figure 7 nanomaterials-12-04316-f007:**
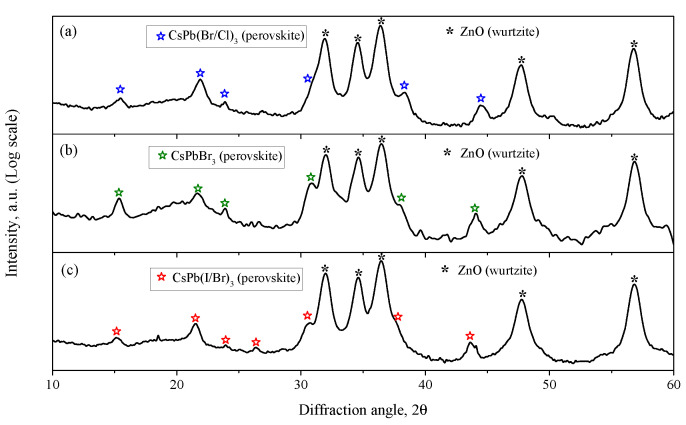
XRD patterns of ZnO/CsPb(Cl/Br)_3_ (**a**), ZnO/CsPbBr_3_ (**b**) and ZnO/CsPb(Br/I)_3_ (**c**) nanocomposites. Due to the large difference in peak intensity between ZnO and PNCs, the intensity is plotted on a logarithmic scale.

**Figure 8 nanomaterials-12-04316-f008:**
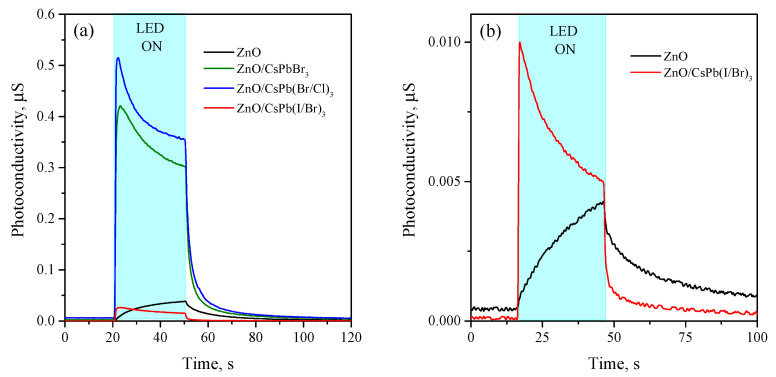
(**a**) Rise and decay curves of photoconductivity of unsensitized ZnO and ZnO/PNCs nanocomposites under blue LED irradiation ($\lambda_{max}=470$ nm, 13 $\mathrm{mW}/{\mathrm{cm}}^{2}$, 30 s) at room temperature; (**b**) Rise and decay curves of photoconductivity of unsensitized ZnO and ZnO/CsPb(I/Br)_3_ nanocomposite under green LED irradiation ($\lambda_{max}=525$ nm, 13 $\mathrm{mW}/{\mathrm{cm}}^{2}$, 30 s) at room temperature.

**Figure 9 nanomaterials-12-04316-f009:**
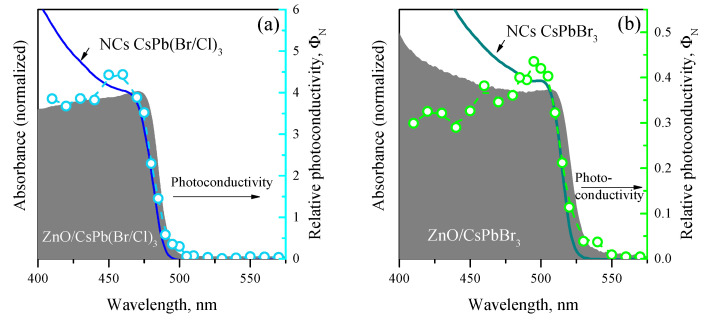
Spectral dependence of photoconductivity of ZnO/CsPb(Cl/Br)_3_ (**a**) and ZnO/CsPbBr_3_ nanocomposites (**b**). The areas with a solid fill relate to the absorption spectra of the corresponding nanocomposites, the solid lines relate to the absorption spectra of the corresponding PNCs hexane dispersions, the dots with a dashed line relate to the photoconductivity spectra of the nanocomposites.

**Figure 10 nanomaterials-12-04316-f010:**
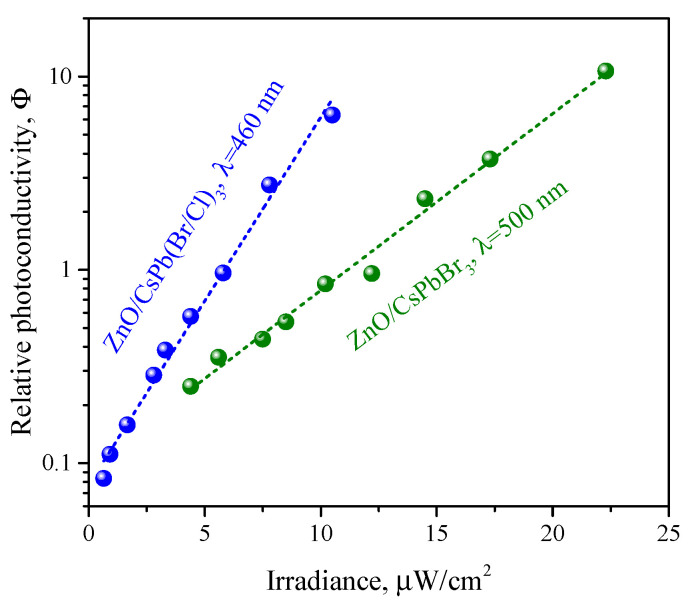
Dependence of photoconductivity on the irradiance for ZnO/CsPb(Cl/Br)_3_ nanocomposite at 460 nm and for ZnO/CsPbBr_3_ nanocomposite at 500 nm.

**Figure 11 nanomaterials-12-04316-f011:**
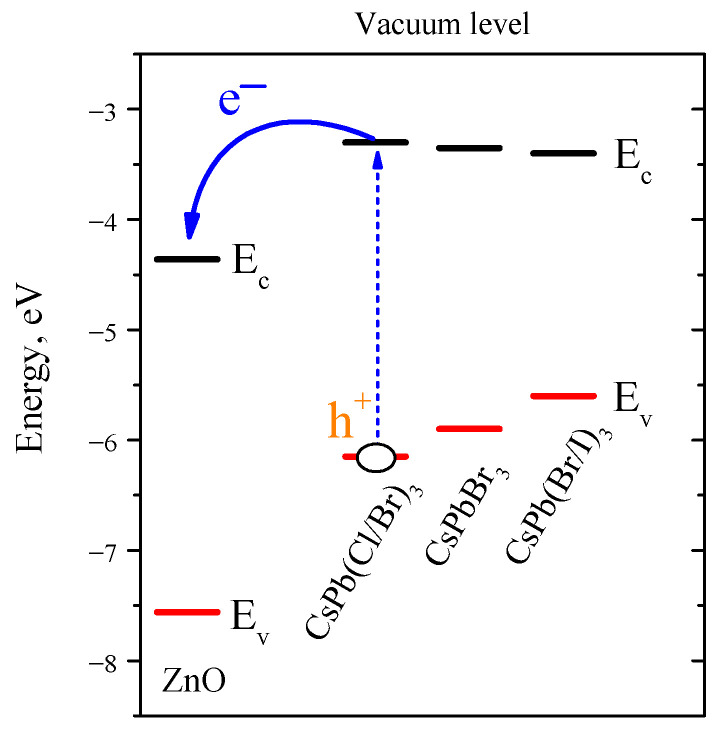
Comparative positions of the edges of the conduction bands ($E_{c}$) and the edges of the valence bands ($E_{v}$) of ZnO and PNCs relative to the vacuum level [66]. Blue arrows show the path of photoinduced electron ($e^{-}$) transfer; the photogenerated hole ($h^{+}$) remains in the valence band of nanocrystals.

**Figure 12 nanomaterials-12-04316-f012:**
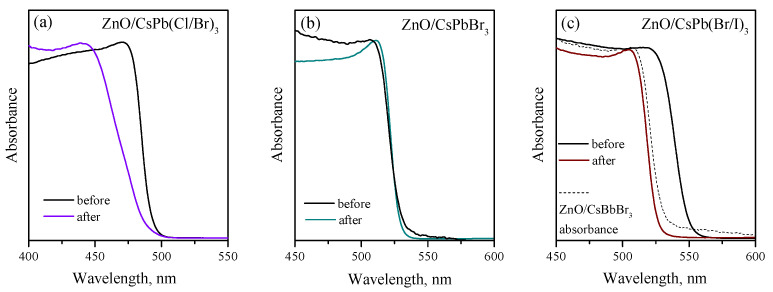
Absorption spectra of ZnO/CsPb(Cl/Br)_3_ (**a**), ZnO/CsPbBr_3_ (**b**) and ZnO/CsPb(Br/I)_3_ (**c**) nanocomposites before and after blue light irradiation (470 nm, 40 $\mathrm{mW}/{\mathrm{cm}}^{2}$, 48 h). On the figure (**c**) a dashed line shows the absorption spectrum of ZnO/CsPbBr_3_ nanocomposite.

**Table 1 nanomaterials-12-04316-t001:** Binding energy of electrons on the core levels of atoms in composition of PNCs.

Sample	$\mathrm{Cs}3d_{5/2}$, eV	$\mathrm{Pb}4f_{7/2}$, eV	$\mathrm{Br}3d_{5/2}$, eV	$\mathrm{Cl}2p_{3/2}$, eV	$I3d_{5/2}$, eV
CsPbBr_3_	724.4(I)	138.4(I)	68.5(I)	-	-
		136.9(II)			
CsPb(Br/Cl)_3_	724.4(I)	138.4(I)	68.5(I)	197.9	-
		136.9(II)			
CsPb(I/Br)_3_	724.0(II)	138.4(I)	68.5(I)	-	618.7(I)
	725.4(III)	139.4(III)	69.3(II)		619.9(II)

**Table 2 nanomaterials-12-04316-t002:** Chemical composition of PNCs in at.% calculated from XP spectra (Figure 6).

Sample	Cs	Pb	Br	Cl	I
CsPbBr_3_	18.1(I)	19.1(I)	60.6(I)	-	-
		2.2(II)			
CsPb(Br/Cl)_3_	14.4(I)	18.9(I)	43.4(I)	18.5	-
		4.8(II)			
CsPb(I/Br)_3_	4.4(II)	9.2(I)	14.8(I)	-	3.6(I)
	10.3(III)	12.5(III)	38.4(II)		6.8(II)

**Table 3 nanomaterials-12-04316-t003:** Photoelectric parameters of ZnO/PNCs nanocomposites and non-sensitized nanocrystalline ZnO under blue LED irradiation ($\lambda_{max}=470$ nm, 13 $\mathrm{mW}/{\mathrm{cm}}^{2}$, 30 s) at room temperature.

Sample	$\sigma_{\mathrm{dark}}$, S, $\,10^{-9}$	On/Off Ratio	R, mA/W	$t_{\mathrm{dec},90\%}$, s
ZnO	0.01	2500	0.7	115
CsPb(Br/Cl)_3_	0.3	1800	20	18
CsPbBr_3_	0.2	2090	16	25
CsPb(I/Br)_3_	0.03	560	0.5	7

## Data Availability

Not applicable.

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
