# Peer review of "Sensitization of ZnO Photoconductivity in the Visible Range by Colloidal Cesium Lead Halide Nanocrystals"

_nanomaterials, 2022, doi:10.3390/nano12234316_

Round 1

Reviewer 1 Report

Please see the comment as attached. 

Author Response

Reviewer #1

  1. As Line 26-30 indicate that Zinc Oxide has band gap of 3.37 eV, then what is the advantage to sensitize Zinc Oxide so that it can be response to visible light comparing with directly using other semiconducting materials?

The author is suggested to polish the introduction portion so that the significance of sensitizing ZnO can be highlighted.

Answer: Thank you for your comment. ZnO has the advantages of high photoconductivity and stability. Other semiconductors with a band gap .directly corresponding to the visible range (metal sulfides, selenides, etc.) exhibit usually lower photoconductivity, and in addition, may be subject to photon-induced degradation due to oxidation. Therefore, there is interest in developing ZnO-based structures with sensitivity to visible light. The following text has been added to the manuscript:

«ZnO exhibits good environmental stability and high photosensitivity, besides being non-toxicity, eco-friendly, earthly abundance and low cost, so the development of ZnO-based photoconductors sensitive to visible light is of of great importance»

  1. The author is suggested to add diagrams of the structures tested in this work. according to this work, the structure should be nanocrystal of CsPbX3 on top of think ZnO films. A diagram can help audience understand it more efficiently.

Answer: Thank you for your comment. The structure diagram is added to the manuscript (Figure 1), with appropriate explanations in the text and figure captions.

Figure 1. Schematic representation of the structure of a ZnO/PNCs photosensitive element, including alumina plates with Pt contacts, a nanocrystalline ZnO thick layer, and PNCs covering the top of ZnO layer.

  1. Figure 3a,d,g is indicating that the number of nanocrystals are also different when changing content of CsPbX3. To make fair comparison between different CsPbX3 sensitized ZnO, should the number of nanocrystals on top of ZnO to be matched so that the electrical or photonic signals are from intrinsic property of the nanocomposite.

Answer: Thank you very much for your comment. Indeed, the number of particles on the graphs is different. This is most likely due to the different “solubility” of nanoparticles in hexane - the larger the size, the worse the solubility and the more dilute the dipersion is formed. Nevertheless, we controlled the concentration of particles in the nanocomposites by the XRF method, and for all of them it was 5±1 wt%. The method for determining the content of perovskite nanocrystals in ZnO-based nanocomposites was previously developed and published by our colleagues, we refer to their article (Ref. 61).

  1. Figure 9 should be corrected with clean plot.

Answer: Thank you very much for your attention. Figure 9 has been modified with clear plot.

  1. Can the author tell why such phenomenon in this work is related to neuromorphic applications, in particular to control the response of artificial optoelectronic synapses, as mentioned in Line 476. It seems to be only the tunning of photoconductivity of ZnO and there is no correlation with optoelectronic synapses. Hope the author can discuss more details in introduction part.

Answer: Thank you very much for your question. Indeed, from our manuscript there is not obvious connection with neuromorphic applications, which should be a continuation of our research. However, we believe that the obtained results are really promising for further neuromorphic applications for a number of reasons: 1) The ability to vary the spectral sensitivity with a change in the composition of perovskites; 2) The unusual behavior of the photoconductivity of nanocomposites upon irradiation, it immediately decreases, in contrast to zinc oxide (and other conventional oxides), in the case of which it gradually increases and reaches saturation. It is interesting to study such behavior, for example, within the framework of such neuromorphic test as pair-pulse facilitation, where both potentiation and depression can be expected in the case of our nanocomposites. In summary, we have added the following sentence to the conclusions:

«ZnO/PNCs nanocomposites demonstrated remarkable photoconductivity kinetics, since their photoconductivity did not rise, but decay during irradiation, which can be used to regulate synaptic plasticity».

Reviewer 2 Report

In this paper authors present their study on the sensitization of  photoconductivity of nanocrystalline ZnO through colloidal perovskite nanocrystals. They sucessfully proved that bromide and mixed chloride-bromide cesium lead halides do effectively sensitize the photoconductivity of ZnO in the spectral range of 400–500 nm. The research was well designed and developed and the results of the experiments made justify well the conclusions. The writing of the paper is at time a bit dense and not so pleasant to read but clear enough. Minor spelling errors and improvements on some phrases throughout the text can be easily done. 

On chapter 2.1.2, first phrase on lines 93 and 94, please consider the write down the "minimal variations" introduced (although their appear later on at the text).

For sake of clarity please re-write the phrase on line 171-173.

Further discussion about the results on graphs b, e and h of figure 3 (lines 201-205) would improve the results analysis.

Good chapter 3.3

Author Response

Reviewer #2

In this paper authors present their study on the sensitization of  photoconductivity of nanocrystalline ZnO through colloidal perovskite nanocrystals. They sucessfully proved that bromide and mixed chloride-bromide cesium lead halides do effectively sensitize the photoconductivity of ZnO in the spectral range of 400-500 nm. The research was well designed and developed and the results of the experiments made justify well the conclusions. The writing of the paper is at time a bit dense and not so pleasant to read but clear enough. Minor spelling errors and improvements on some phrases throughout the text can be easily done.

Answer: Thank you for the positive evaluation of our work. We have made English corrections to the manuscript where errors were found.

  1. On chapter 2.1.2, first phrase on lines 93 and 94, please consider the write down the "minimal variations" introduced (although their appear later on at the text).

Answer: Thanks for your remark. The following phrase has been added to the text, disclosing the minimal variations that have been made:

(Argon (99.998 %, 7~ppm oxygen and 9~ppm water vapor) was used as an inert gas, the amounts of reagents were doubled compared to the original method)

  1. For sake of clarity please re-write the phrase on line 171-173.

Answer: Thank you for your comment. This phrase has been rewritten as follows:

The photoconductive elements were fixed in a gas- and lightproof Teflon cell. A LED (λmax=470 nm) inside the cell was used for irradiation. The distance from the LED to the photoconductive elements was approximately 4 cm.

  1. Further discussion about the results on graphs b, e and h of figure 3 (lines 201-205) would improve the results analysis.

Thank you very much for your comment. The following text has been added to the manuscript in order of discussion:

«An increase in polydispersity in the case of iodine-containing PNCs may be due to a higher rate of the reaction of formation of nanoparticles, as a result of which the growth of nanoparticles begins immediately after the injection of cesium oleate, even before the full mixing of precursors. On the other hand, during the synthesis of chlorine-containing PNCs, the growth of nanocrystals is slowed down and begins a few seconds after the injection of cesium oleate. Thus, the precursors well mixed and conditions are created for uniform supersaturation throughout the volume of the reaction mixture, resulting in the formation of monodisperse nanoparticles.»

  1. Good chapter 3.3

Answer: Thank you for the positive evaluation of this section.
